# Mediterranean Diet Decreases the Initiation of Use of Vitamin K Epoxide Reductase Inhibitors and Their Associated Cardiovascular Risk: A Randomized Controlled Trial

**DOI:** 10.3390/nu12123895

**Published:** 2020-12-19

**Authors:** Sara Castro-Barquero, Margarita Ribó-Coll, Camille Lassale, Anna Tresserra-Rimbau, Olga Castañer, Xavier Pintó, Miguel Ángel Martínez-González, José V. Sorlí, Jordi Salas-Salvadó, José Lapetra, Enrique Gómez-Gracia, Ángel M. Alonso-Gómez, Miquel Fiol, Lluis Serra-Majem, Emilio Sacanella, Francisco Javier Basterra-Gortari, Olga Portolés, Nancy Babio, Montserrat Cofán, Emilio Ros, Ramón Estruch, Álvaro Hernáez

**Affiliations:** 1Cardiovascular Risk, Nutrition and Aging Research Group, August Pi i Sunyer Biomedical Research Institute (IDIBAPS), 08036 Barcelona, Spain; sacastro@clinic.cat (S.C.-B.); mribocoll@gmail.com (M.R.-C.); esacane@clinic.cat (E.S.); restruch@clinic.cat (R.E.); 2Department of Medicine, Faculty of Medicine and Health Sciences, University of Barcelona, 08036 Barcelona, Spain; 3Consorcio CIBER, M.P. Fisiopatología de la Obesidad y Nutrición (CIBEROBN), Instituto de Salud Carlos III, 28029 Madrid, Spain; classale@imim.es (C.L.); annatresserra@ub.edu (A.T.-R.); ocastaner@imim.es (O.C.); xpinto@bellvitgehospital.cat (X.P.); mamartinez@unav.es (M.Á.M.-G.); Sorli@uv.es (J.V.S.); jordi.salas@urv.cat (J.S.-S.); joselapetra543@gmail.com (J.L.); egomezgracia@uma.es (E.G.-G.); angelmago13@gmail.com (Á.M.A.-G.); miguel.fiol@ssib.es (M.F.); lserra@dcc.ulpgc.es (L.S.-M.); javierbasterra@hotmail.com (F.J.B.-G.); olga.portoles@uv.es (O.P.); nancy.babio@urv.cat (N.B.); mcofan@clinic.cat (M.C.); eros@clinic.cat (E.R.); 4Faculty of Pharmacy and Food Science, Universitat de Barcelona, 08028 Barcelona, Spain; 5Cardiovascular Risk and Nutrition Research Group, Hospital del Mar Medical Research Institute (IMIM), 08003 Barcelona, Spain; 6Unitat de Nutrició Humana, Departament de Bioquimica i Biotecnologia, Hospital Universitari Sant Joan de Reus, Universitat Rovira i Virgili, 43201 Reus, Spain; 7Institut d′Investigació Pere Virgili (IISPV), 43204 Reus, Spain; 8Department of Nutrition, Food Science and Gastronomy, XaRTA, INSA, Faculty of Pharmacy and Food Sciences, University of Barcelona, 08028 Barcelona, Spain; 9Lipids and Vascular Risk Unit, Internal Medicine Service, Hospital Universitario de Bellvitge, 08907 L’Hospitalet de Llobregat, Spain; 10Department of Preventive Medicine and Public Health, Universidad de Navarra, 31008 Pamplona, Spain; 11Department of Nutrition, Harvard TH Chan School of Public Health, Boston, MA 02115, USA; 12Department of Preventive Medicine, Universidad de Valencia, 46100 Valencia, Spain; 13Department of Family Medicine, Research Unit, Distrito Sanitario Atención Primaria Sevilla, 41013 Sevilla, Spain; 14Department of Preventive Medicine and Public Health, Universidad de Málaga, 29071 Málaga, Spain; 15Bioaraba Health Research Institute, Osakidetza Basque Health Service, Araba University Hospital, University of the Basque Country UPV/EHU, 01009 Vitoria-Gasteiz, Spain; 16Health Research Institute of the Balearic Islands (IdISBa), Hospital Son Espases, 07120 Palma de Mallorca, Spain; 17Instituto de Investigaciones Biomédicas y Sanitarias, Universidad de Las Palmas de Gran Canaria, 35016 Las Palmas, Spain; 18Centro Hospitalario Universitario Insular Materno Infantil (CHUIMI), Servicio Canario de Salud, 35016 Las Palmas, Spain; 19Internal Medicine Service, Hospital Clínic, 08036 Barcelona, Spain; 20Department of Endocrinology and Nutrition, Complejo Hospitalario de Navarra, 31008 Pamplona, Spain; 21Lipid Clinic, Endocrinology and Nutrition Service, Hospital Clínic, 08036 Barcelona, Spain; 22Blanquerna School of Health Sciences, Universitat Ramon Llull, 08025 Barcelona, Spain

**Keywords:** Mediterranean Diet, randomized controlled trials, prevention, 4-hydroxycoumarins, platelet aggregation inhibitors

## Abstract

Our aim is to assess whether following a Mediterranean Diet (MedDiet) decreases the risk of initiating antithrombotic therapies and the cardiovascular risk associated with its use in older individuals at high cardiovascular risk. We evaluate whether participants of the *PREvención con DIeta MEDiterránea* (PREDIMED) study allocated to a MedDiet enriched in extra-virgin olive oil or nuts (versus a low-fat control intervention) disclose differences in the risk of initiation of: (1) vitamin K epoxide reductase inhibitors (acenocumarol/warfarin; *n* = 6772); (2) acetylsalicylic acid as antiplatelet agent (*n* = 5662); and (3) other antiplatelet drugs (cilostazol/clopidogrel/dipyridamole/ditazol/ticlopidine/triflusal; *n* = 6768). We also assess whether MedDiet modifies the association between the antithrombotic drug baseline use and incident cardiovascular events. The MedDiet intervention enriched with extra-virgin olive oil decreased the risk of initiating the use of vitamin K epoxide reductase inhibitors relative to control diet (HR: 0.68 [0.46–0.998]). Their use was also more strongly associated with an increased risk of cardiovascular disease in participants not allocated to MedDiet interventions (HR_control diet_: 4.22 [1.92–9.30], HR_MedDiets_: 1.71 [0.83–3.52], *p*-interaction = 0.052). In conclusion, in an older population at high cardiovascular risk, following a MedDiet decreases the initiation of antithrombotic therapies and the risk of suffering major cardiovascular events among users of vitamin K epoxide reductase inhibitors.

## 1. Introduction

Compelling evidence from observational studies and randomized controlled trials such as the PREDIMED (*PREvención con DIeta MEDiterránea*) study indicates that following a Mediterranean Diet (MedDiet) decreases the risk of incident cardiovascular disease [1,2,3]. It is believed that the MedDiet exerts these benefits via improving glucose metabolism, endothelial function, oxidative stress, and low-grade inflammation [4], but little is known about the effects of this dietary pattern on thrombosis-related outcomes. Following a MedDiet has been associated with improvements in atherothrombosis biomarkers [5], platelet function [6], and platelet count [7], as well as with decreases in the circulating levels of pro-thrombotic microvesicles [8]. However, no intervention trial has assessed the effects of a healthy dietary pattern like the MedDiet on more clinical outcomes related to thrombosis such as the initiation of antithrombotic medications. A reduction in the risk of initiating the use of vitamin K epoxide reductase inhibitors (molecules preventing the in vivo regeneration of vitamin K, necessary for the activation—through carboxylation—of coagulation factors II, VII, IX, and X which, in turn, decrease the capacity of the organism to trigger the coagulation cascade) could be due to a decreased incidence of pathologies for which they are the treatment of choice and on which MedDiet has not yet shown a beneficial effect, such as deep vein thrombosis [9]. In addition, a decline in the use of antiplatelet drugs could be related to a reduction in the presence of individuals with severe atherosclerotic lesions [10]. Besides, no study has shown whether following a MedDiet decreases cardiovascular risk in populations treated with any of these medications.

The aims of our study are to assess whether following a MedDiet: (1) decreases the risk of initiating antithrombotic drugs in non-users; and (2) modulates the association between the use of antithrombotic drugs at the baseline and the risk of developing a major cardiovascular event.

## 2. Materials and Methods

### 2.1. Study Population

The study subjects participated in the PREDIMED study. It was a multicenter, randomized, controlled trial conducted in Spain aiming to assess the effects of following a MedDiet on the primary prevention of cardiovascular outcomes in high cardiovascular risk patients. Eligible participants were men (aged 55–80 years) and women (aged 60–80 years) free of cardiovascular disease at enrolment but diagnosed with type 2 diabetes or presenting three or more of the following cardiovascular risk factors: smoking, hypertension, high concentrations of low-density lipoprotein cholesterol, low concentrations of high-density lipoprotein cholesterol, overweight/obesity, and family history of premature coronary heart disease. Enrollment began on 25 June 2003, and the last participant was included on 30 June 2009. The PREDIMED trial protocol complied with the Declaration of Helsinki, was approved by Institutional Review Boards of all recruiting centers, was registered under the International Standard Randomized Controlled Trial Number ISRCTN35739639 (http://www.isrctn.com/ISRCTN35739639), and is available in the study website (http://www.predimed.es). All participants provided written informed consent before joining the trial (a blank informed consent form of the study is provided in Appendix A). An institutional ethics committee (CEIC-PSMAR) approved the particular protocol of this sub-project (code: 2018/8180/I, date: 4 December 2018). More information on recruiting methods, inclusion/exclusion criteria, and data collection has been described in detail elsewhere [1,3].

Of the total 7447 randomized participants in the PREDIMED trial, we excluded 87 with no available data on baseline MedDiet adherence or alcohol consumption. We assessed the initiation of vitamin K epoxide reductase inhibitors, acetylsalicylic acid, and non-acetylsalicylic acid antiplatelet drugs in individuals with follow-up information on drug use (we excluded 297 participants because of this) and not using these medications at the baseline (we, respectively, excluded the 135, 1136, and 124 users of vitamin K epoxide reductase inhibitors, acetylsalicylic acid, and other antiplatelet drugs at the baseline). Users of non-acetylsalicylic acid antiplatelet drugs and vitamin K epoxide reductase inhibitors at the baseline were additionally excluded from the analyses on the initiation of use of vitamin K epoxide reductase inhibitors and non-acetylsalicylic acid antiplatelet drugs, respectively. Finally, as explained in the Statistical Methods section, any initiation in drug use during the first year of follow-up was excluded to minimize reverse causation. The study flowchart is available in Figure 1. The CONSORT checklist of our study is available in Appendix A.

### 2.2. Dietary Intervention

Three intervention arms (allocation ratio 1:1:1) were compared: (1) a MedDiet enriched with extra-virgin olive oil (MedDiet-EVOO); (2) a MedDiet enriched with nuts (MedDiet-Nuts); and (3) a low-fat control diet. MedDiet interventions promoted: (1) the consumption of fruits, vegetables, pulses, mixed nuts, and fish; (2) the use of extra-virgin olive oil as main culinary fat and the use of traditional culinary preparations such as “sofrito”; (3) the substitution of red and processed meat for poultry; and (4) a decrease in the intake of spreadable fats, fried snacks, sugary soft drinks, commercial pastries, bakery goods, and sweets. Volunteers allocated to the MedDiet-EVOO intervention were given 1 L/week of virgin olive oil and those in the MedDiet-Nuts group were provided with 210 g/week of mixed nuts to promote compliance and account for family needs. Volunteers allocated to the low-fat control group were recommended: (1) to promote the consumption of fruits, vegetables, pulses, lean fish, and low-fat dairy products; (2) to decrease the intake of vegetable oils (including olive oils), “sofrito”, nuts, red and processed meat, visible fat in meats and other recipes, fatty fish, seafood canned in oil, spreadable fats, fried snacks, sugary soft drinks, commercial pastries, bakery goods, and sweets. It must be pointed out that olive oil is a remarkable dietary source of vitamin K1 (phylloquinone), providing 60.2 μg/100 g of oil (which implies that each 10 g spoonful of oil provides 5.0 and 6.8% of the daily Adequate Intakes of vitamin K in adult men and women, respectively, according to the Institute of Medicine) [11,12]. Further details of the dietary protocol have been described in detail [1,3].

### 2.3. Outcomes

In the annual study visits we collected data on the use (yes/no) of the main families of antithrombotic drugs: (1) vitamin K epoxide reductase inhibitors (warfarin and acenocumarol); (2) acetylsalicylic acid used as antiplatelet agent; and (3) other antiplatelet therapies (other cyclooxygenase inhibitors –triflusal, ditazol–, glycoprotein IIb/IIIa P2Y12 platelet receptor antagonists –clopidogrel, ticlopidine–, and modulators of cyclic adenosine monophosphate metabolism in platelets –dipyridamole, cilostazol–). Using these data, we calculated incidence and time-to-event of new use of any of these three drug classes among baseline non-users. We defined “onset” as the appearance of the use of any of the above medications during a follow-up period that lasted until the last visit of the volunteer [13]. Regarding doubtful cases, we only considered as valid outcomes any start of drug use that persisted for at least three subsequent follow-up visits and was not based on more than one visit in which the use of the drug was not reported.

For our secondary analyses, we collected information on the development of non-fatal coronary heart disease, non-fatal stroke, and death from these causes. Incidence up to 1 December 2010 and time-to-event values of these conditions were determined by the Clinical Event Committee through follow-up visits, repeated contact with participants, yearly review of medical records between 2011 and 2017, and linkage with the national death registry [1,3]. Information on incident events of atrial fibrillation was also collected following the same methodology [14].

### 2.4. Clinical and Lifestyle Vovariates

Trained personnel collected baseline data on age; sex; educational level; prevalence of diabetes, hypercholesterolemia, hypertriglyceridemia, and hypertension; body mass index; and smoking habit. We estimated baseline alcohol intake (in g/day) using a 137-item, semi-quantitative food frequency questionnaire validated in Spanish adults [15]. Finally, we also estimated leisure-time physical activity levels (in metabolic equivalents of task-minute per day) by a Minnesota Leisure-Time Physical Activity Questionnaire validated in Spanish adult men and women [16,17].

### 2.5. Power Analysis

The number of total individuals included in each analysis and the number of cases that occurred during follow-up allowed ≥80% power to detect as significant (*p*-value < 0.05) hazard ratios (HR) for the comparisons between control diet and MedDiet-EVOO or MedDiet-Nuts, respectively, of the following values: ≤0.57 and ≤0.58 (initiation of vitamin K epoxide reductase inhibitor use), ≤0.77 and ≤0.76 (initiation of acetylsalicylic acid use), and ≤0.55 and ≤0.55 (initiation of other antiplatelet therapies) (Appendix A). We performed these analyses using the “powerSurvEpi” package in R Software [18].

### 2.6. Statistical Analyses

We defined the baseline traits of the participants by means and standard deviation (normally distributed continuous variables), medians and interquartile range (non-normally distributed continuous variables), and proportions (categorical variables).

We assessed the differences in the risk of initiating the use of vitamin K epoxide reductase inhibitors, acetylsalicylic acid as antiplatelet agent, and other antiplatelet drugs by three sets of Cox regression models. We defined follow-up time as the time between the date of enrollment and: (1) the midpoint between the last visit in which the participant did not use the medication of interest and the first visit in which he/she used it [19]; (2) 6 years of maximum follow-up time [13]; or (3) 1 December 2010, whichever came first. Cases of first use of antithrombotic drugs registered at the first follow-up visit were excluded to avoid reverse causation. We estimated HRs in the two MedDiet intervention groups relative to the control diet and fitted two models. Model 1 was adjusted for sex and recruitment site as strata variables, and age (continuous). Model 2 was further adjusted for educational level (primary/secondary/higher/unavailable, as strata variable), diabetes (yes/no), hypercholesterolemia (yes/no), hypertriglyceridemia (yes/no), hypertension (yes/no), smoking habit (current/former/never), leisure-time physical activity (continuous), body mass index (continuous), alcohol consumption (continuous), and two propensity scores that used 30 baseline variables to estimate the probability of assignment to each of the intervention groups [1]. Following the strategy described in previous publications of the PREDIMED Study [1], we used robust variance estimators to account for intra-cluster correlations in both models. To study mid-term effects of the intervention, we assessed the differences in the incidence of the outcomes during the first 4 years of the study following the same methodology. We also displayed incident cases in the three intervention groups using Kaplan–Meier cumulative incidence curves (weighted by inverse probability weighting according to a propensity score model of assignment to MedDiet intervention or control group based on the covariates above listed).

Our second aim was to determine whether adherence to the MedDiet modified the association between antithrombotic drug use at the baseline and the incidence of major adverse cardiovascular events. We compared the volunteers allocated to the MedDiet interventions relative to the control group. We fitted Cox models where the outcome was the occurrence of the first major cardiovascular event and included an interaction product-term “antithrombotic drug use x group”. We applied a likelihood ratio test between the models with and without the interaction term to assess whether the interaction was significant. We adjusted for the covariates in the main objective. In addition, we aimed to minimize indication bias by adjusting for propensity scores that estimated the probability of being user of the drug at the baseline (calculated according to the covariates of the model) [20] and used robust variance estimators.

We fitted Cox models using the “survival” package in R Software (version 3.5.2) [21,22].

## 3. Results

### 3.1. Participants

For this study, we studied the risk of initiation of vitamin K epoxide reductase inhibitors, acetylsalicylic acid, and other antiplatelet agents in participants not using them at the baseline (*n* = 6772, *n* = 5662, and *n* = 6768, respectively). Participants were older adults (67 years old on average, 58–59% women) with a high prevalence of cardiovascular risk factors (82–83% hypertension, 72% hypercholesterolemia, 44–49% diabetes, 47% obesity, 29% hypertriglyceridemia, 14% current smokers) (Table 1). Median follow-up time was 4.5, 4.0, and 4.6 years for the assessment of the risk of new users of vitamin K epoxide reductase inhibitors, acetylsalicylic acid, and other antiplatelet drugs, respectively.

### 3.2. MedDiet Effects on the Initiation of Antithrombotic Therapy

The risk of becoming a new vitamin K epoxide reductase inhibitor user was 32% lower in the MedDiet-EVOO intervention compared to the control diet group (HR: 0.68 [95% confidence interval: 0.46–0.998], *p* = 0.049) (Table 2). The difference between the incidence rate in the control diet and the MedDiet-EVOO intervention was 0.66%. After excluding incident cases of atrial fibrillation from the analysis (due to their almost universal requirement of vitamin K epoxide reductase inhibitors), following the MedDiet-EVOO intervention decreased the initiation risk by 47% (HR: 0.53 [95% CI: 0.32; 0.88], *p* = 0.014). MedDiet interventions had no effects on the incidence of new users of acetylsalicylic acid (Table 3). Finally, regarding other antiplatelet drugs, a mid-term effect in the MedDiet-EVOO intervention was suggested (Figure 2B). When we restricted the analyses to a maximum follow-up time of 4 years, the risk of initiating non-acetylsalicylic acid antiplatelet therapy was reduced in this intervention arm (in the model adjusted for age, sex, and recruitment site, HR: 0.60 [95% CI: 0.36–0.99], *p* = 0.045; in the model further adjusted, HR: 0.62 [95% CI: 0.38–1.04], *p* = 0.069) (Table 4).

### 3.3. Interaction between Antithrombotic Therapy at the Baseline and MedDiet on the Incidence of Cardiovascular Events

Use of vitamin K epoxide reductase inhibitors at the baseline was associated with 153% higher risk of suffering a major adverse cardiovascular event (HR: 2.53 [95% CI: 1.52; 4.19], *p* < 0.001). However, this association was much stronger for participants in the control group and blunted in those allocated to the MedDiets (HR_control diet_: 4.22 [95% CI: 1.92; 9.30], *p* < 0.001; HR_MedDiet_: 1.71 [95% CI: 0.83; 3.52], *p* = 0.137; *p*-interaction = 0.052) (Figure 3A). Acetylsalicylic acid use at the baseline was unrelated to greater cardiovascular risk (Figure 3B). Finally, use of non-acetylsalicylic acid antiplatelet drug was linked to greater cardiovascular risk (HR: 1.93 [95% CI: 1.02; 3.66], *p* = 0.045), independently from the allocation to any intervention group (Figure 3C). Exact values are available in Appendix A.

## 4. Discussion

Our results indicate that following a MedDiet enriched in extra-virgin olive oil reduced the risk of initiating treatment with vitamin K epoxide reductase inhibitors in older individuals at high cardiovascular risk. A 4-year decrease in the risk of starting to use non-acetylsalicylic acid antiplatelet drugs is also suggested. Likewise, MedDiet attenuated the association between the use of vitamin K epoxide reductase inhibitors and a greater risk of suffering a major cardiovascular event. 

Vitamin K epoxide reductase inhibitors are one of the basic pharmacological strategies against atrial fibrillation, venous thromboembolism, and thrombotic responses after surgical interventions [9]. Following a MedDiet has been shown to be able to decrease the incidence of diverse atherosclerotic cardiovascular events in observational and interventional studies [1,2]. Our findings suggest that its effects could be extended to other thrombosis-related outcomes, since the MedDiet-EVOO decreased the risk of initiating vitamin K epoxide reductase inhibitor use. This effect is independent from the association between the MedDiet-EVOO intervention and lower risk of developing atrial fibrillation in high cardiovascular risk individuals [14] (one of the main indications of this pharmacological therapy), suggesting a benefit on a proxy of incidence of thrombosis-related diseases. Our findings also indicate that following a MedDiet attenuated the association of using vitamin K epoxide reductase inhibitors with greater risk of suffering a major cardiovascular event, reporting a protective effect for treated and non-treated individuals. The role of vitamin K in cardiovascular mechanisms and the intake of vitamin K within a frail population following a MedDiet has been previously discussed [23] but, to the best of our knowledge, no study to date had assessed the protective effect of this dietary pattern on vitamin K-related medications.

Non-acetylsalicylic acid antiplatelet drugs have been shown to be an effective protective pharmacological strategy for the primary prevention of non-fatal atherosclerotic diseases [24]. Our findings suggest a decrease in the mid-term initiation of this therapy in the MedDiet-EVOO intervention group. We have also observed that baseline use of these drugs in our population was associated with an increased risk of incident atherosclerotic disease, which suggests that these therapies have been prescribed to participants more likely to suffer a cardiovascular outcome. The fact that MedDiet is related to lower incidence of atherosclerotic disease in the PREDIMED study [1,25], as well as in other intervention trials and observational studies [2], may explain a subsequent decrease in the use of these medications.

In order to explain MedDiet benefits on antithrombotic drug use, it should be considered that thrombosis responses are strongly modulated by inflammation and oxidative stress [26]. MedDiet is known to improve these risk factors [4,27,28], probably explaining its beneficial effects on thrombosis as well. Three dietary components of the MedDiet pattern may act synergistically to moderate thrombotic responses. First, antioxidants (in extra-virgin olive oil, fruits, vegetables, legumes, and nuts) neutralize reactive species of oxygen and nitrogen and decrease their capacity to promote platelet activation and stimulate the coagulation cascade [29]. Antioxidants may also increase the half-life and bioavailability of nitric oxide (capable of inhibiting excessive platelet activation) by decreasing its transformation into peroxynitrite (these compounds neutralize superoxide anions, which react with nitric oxide generating peroxynitrite) [30]. Second, omega-3 polyunsaturated fatty acids (in fish, seafood, and nuts) are transformed into antithrombotic eicosanoids such as 3-series prostaglandins and thromboxanes and 5-series leukotrienes [31]. Finally, short-chain fatty acids (such as butyric, propionic, and acetic acids; generated by the fermentation of dietary fiber by probiotic bacteria in the intestine) and some phenolic compounds could contribute to this protection. In particular, these molecules stimulate adenosine monophosphate-activated protein kinase [32,33], a metabolic regulator able to induce the production of nitric oxide (via the activation of nitric oxide synthases) and the synthesis of antioxidant and anti-inflammatory enzymes [34].

Our study has limitations. First, onset of antithrombotic medication was not a predetermined endpoint in the PREDIMED study, therefore, these analyses should be considered as exploratory. Second, the use of more modern antithrombotic therapies (e.g., direct oral anticoagulants such as dabigatran, apixaban, rivaroxaban, and edoxaban) was extremely scarce during our study follow-up (2003–2010) and we were unable to investigate changes in the risk of initiating the use of these medications. Third, the study sample (older individuals at high cardiovascular risk) limits the generalizability of the results to other populations. Fourth, some covariates in our analyses such as leisure-time physical activity were self-reported and this may imply some residual confounding. Fifth, regarding antithrombotic treatment, we could only collect categorical information on drug use/non-use and, consequently, we could not assess dose changes in treated individuals. Sixth, we were only able to report moderate effects on the outcomes of interest, considering that our intervention was based on modest real-life dietary modifications and that the control diet was already a healthy, low-fat dietary pattern. Finally, the *p*-value for the interaction between the use of vitamin K epoxide reductase inhibitors at the baseline and the intervention group on the risk of developing a major adverse cardiovascular event was only marginally significant (*p* = 0.052). However, we decided to interpret the interaction since the association between drug use and cardiovascular incidence was very strong in the participants allocated to the control group but non-significant (and of a very lower magnitude) in those allocated to MedDiet, and several previous publications have allowed the consideration of marginally significant *p*-interaction values (*p* < 0.1) due to the extremely demanding nature of this type of analyses [35,36].

## 5. Conclusions

Our findings suggest that following a MedDiet in an older population at high cardiovascular risk may reduce the initiation of vitamin K epoxide reductase inhibitor use and non-acetylsalicylic acid antiplatelet therapy, and decrease the risk of suffering major adverse cardiovascular events among vitamin K epoxide reductase inhibitor users. Our results suggest that adherence to the MedDiet could be a beneficial strategy in high cardiovascular risk individuals prone to develop thrombotic outcomes.

## Figures and Tables

**Figure 1 nutrients-12-03895-f001:**
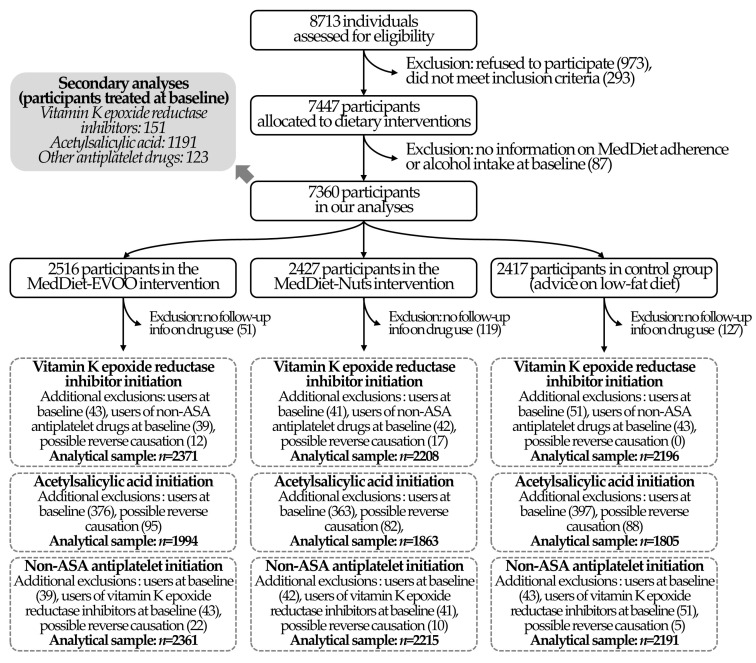
Flow chart of the study.

**Figure 2 nutrients-12-03895-f002:**
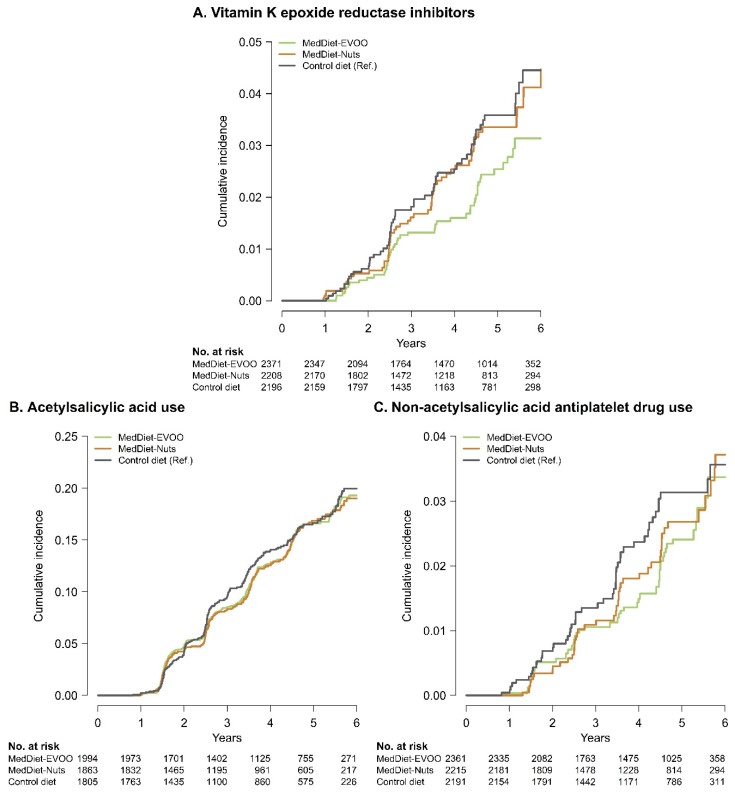
Incident cases of initiation of use of vitamin K epoxide reductase inhibitors (**A**), acetylsalicylic acid as antiplatelet agent (**B**), and non-acetylsalicylic antiplatelet drugs (**C**) in the three intervention groups by Kaplan–Meier cumulative incidence curves. Kaplan–Meier curves weighted by inverse probability weighting using a propensity score model of assignment to intervention or control group based on: age, sex, recruitment site, educational level, diabetes, hypercholesterolemia, hypertriglyceridemia, smoking, leisure-time physical activity, body mass index, alcohol consumption, and two propensity scores that used 30 baseline variables to estimate the probability of assignment to each of the intervention groups. MedDiet-EVOO: Mediterranean Diet enriched with extra-virgin olive oil; MedDiet-Nuts: Mediterranean Diet enriched with nuts.

**Figure 3 nutrients-12-03895-f003:**
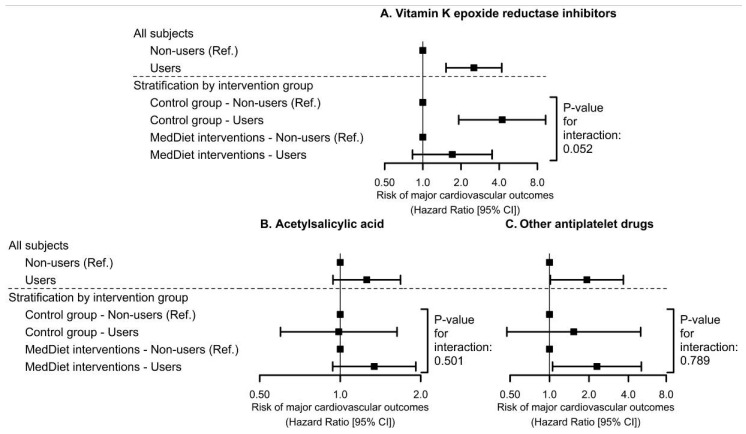
Associations of baseline use of vitamin K epoxide reductase inhibitors (**A**), acetylsalicylic acid as antiplatelet agent (**B**), and non-acetylsalicylic antiplatelet drugs (**C**) with the risk of suffering a major cardiovascular event stratified by intervention group. Hazard ratios were estimated by multivariable Cox proportional hazards regression models adjusted for sex, recruitment site, educational level, diabetes, hypercholesterolemia, hypertriglyceridemia, hypertension, smoking habit, leisure-time physical activity, body mass index, alcohol consumption (at baseline), and two propensity scores that used 30 baseline variables to estimate the probability of assignment to each of the intervention groups. We used robust standard errors to account for intra-cluster correlations. MedDiet: Mediterranean Diet.

**Table 1 nutrients-12-03895-t001:** Study population groups.

	Non-Users of Vitamin K Epoxide Reductase Inhibitors at Baseline (*n* = 6772)	Non-Users of Acetylsalicylic Acid at Baseline (*n* = 5662)	Non-Users of Non-Acetylsalicylic Acid Antiplatelet Drugs at Baseline (*n* = 6768)
Age (years), mean ± SD	66.9 ± 6.16	66.8 ± 6.16	66.9 ± 6.16
Female sex, *n* (%)	3939 (58.2)	3344 (59.1)	3938 (58.2)
Diabetes, *n* (%)	3289 (48.6)	2498 (44.1)	3279 (48.4)
Hypercholesterolemia, *n* (%)	4894 (72.3)	4063 (71.8)	4891 (72.3)
Hypertriglyceridemia, *n* (%)	1955 (28.9)	1613 (28.5)	1951 (28.8)
Hypertension, *n* (%)	5585 (82.5)	4688 (82.8)	5583 (82.5)
Smoking habit:			
Never smokers, *n* (%)	4176 (61.7)	3519 (62.2)	4180 (61.8)
Current smokers, *n* (%)	951 (14.0)	799 (14.1)	948 (14.0)
Former smokers, *n* (%)	1645 (24.3)	1344 (23.7)	1640 (24.2)
Weight status (according to body mass index):			
18.5–24.9 kg/m^2^, *n* (%)	500 (7.38)	393 (6.94)	498 (7.36)
25.0–29.9 kg/m^2^, *n* (%)	3074 (45.4)	2631 (46.5)	3072 (45.4)
≥30.0 kg/m^2^, *n* (%)	3198 (47.2)	2638 (46.6)	3198 (47.3)
PREDIMED Intervention groups:			
MedDiet-EVOO, *n* (%)	2369 (35.0)	1994 (35.2)	2361 (34.9)
MedDiet-Nuts, *n* (%)	2208 (32.6)	1863 (32.9)	2215 (32.7)
Low-fat control diet, *n* (%)	2195 (32.4)	1805 (31.9)	2192 (32.4)
Leisure-time physical activity (metabolic equivalents of task-minute/day), median (1st–3rd quartile)	175 (67.8–319)	175 (65.1–319)	175 (66.2–319)

MedDiet-EVOO: Mediterranean Diet enriched with extra-virgin olive oil; MedDiet-Nuts: Mediterranean Diet enriched with mixed nuts.

**Table 2 nutrients-12-03895-t002:** Incidence of new users of vitamin K epoxide reductase inhibitors

	Cases/Total (Incidence Rate)	Model 1 (HR [95% CI])	Model 2 (HR [95% CI])	Cases/Total (Incidence Rate)	Model 1 (HR [95% CI])	Model 2 (HR [95% CI])
	Vitamin K Epoxide Reductase Inhibitors			Vitamin K Epoxide Reductase Inhibitors Excluding Cases of Atrial Fibrillation		
Control diet	58/2196 (2.64%)	1 (Ref.)	1 (Ref.)	36/2139 (1.68%)	1 (Ref.)	1 (Ref.)
MedDiet-EVOO	47/2371 (1.98%)	0.64 [0.43; 0.94] (*p* = 0.021)	0.68 [0.46; 0.998] (*p* = 0.049)	23/2316 (0.99%)	0.50 [0.30; 0.83] (*p* = 0.008)	0.53 [0.32; 0.88] (*p* = 0.014)
MedDiet-Nuts	56/2208 (2.54%)	0.90 [0.62; 1.30] (*p* = 0.559)	0.97 [0.67; 1.41] (*p* = 0.886)	33/2155 (1.53%)	0.86 [0.53; 1.38] (*p* = 0.526)	0.95 [0.60; 1.51] (*p* = 0.829)

Hazard ratios were estimated by multivariable Cox proportional hazards regression models. Model 1 was adjusted for sex and recruitment site as strata variables, and age. Model 2 was further stratified by educational level as strata variable, diabetes, hypercholesterolemia, hypertriglyceridemia, hypertension, smoking habit, leisure-time physical activity, body mass index, alcohol consumption (at baseline); and two propensity scores that used 30 baseline variables to estimate the probability of assignment to each of the intervention groups. We used robust standard errors to account for intra-cluster correlations. MedDiet-EVOO: Mediterranean Diet enriched with extra-virgin olive oil; MedDiet-Nuts: Mediterranean Diet enriched with mixed nuts.

**Table 3 nutrients-12-03895-t003:** Incidence of new users of acetylsalicylic acid as antiplatelet drug.

	Cases/Total (Incidence Rate)	Model 1 (HR [95% CI])	Model 2 (HR [95% CI])
	Acetylsalicylic Acid as Antiplatelet Drug		
Control diet	223/1805 (12.4%)	1 (Ref.)	1 (Ref.)
MedDiet-EVOO	278/1994 (13.9%)	0.98 [0.82; 1.17] (*p* = 0.800)	0.98 [0.82; 1.18] (*p* = 0.833)
MedDiet-Nuts	231/1863 (12.4%)	0.96 [0.80; 1.16] (*p* = 0.668)	1.00 [0.83; 1.20] (*p* = 0.977)

Hazard ratios were estimated by multivariable Cox proportional hazards regression models. Model 1 was adjusted for sex and recruitment site as strata variables, and age. Model 2 was further stratified by educational level as strata variable, diabetes, hypercholesterolemia, hypertriglyceridemia, hypertension, smoking habit, leisure-time physical activity, body mass index, alcohol consumption (at baseline), and two propensity scores that used 30 baseline variables to estimate the probability of assignment to each of the intervention groups. We used robust standard errors to account for intra-cluster correlations. MedDiet-EVOO: Mediterranean Diet enriched with extra-virgin olive oil; MedDiet-Nuts: Mediterranean Diet enriched with mixed nuts.

**Table 4 nutrients-12-03895-t004:** Incidence of new users of non-acetylsalicylic acid antiplatelet drugs.

	Cases/Total (Incidence Rate)	Model 1 (HR [95% CI])	Model 2 (HR [95% CI])	Cases/Total (Incidence Rate)	Model 1 (HR [95% CI])	Model 2 (HR [95% CI])
	Non-Acetylsalicylic Acid Antiplatelet Drugs (4 Years of Maximal Follow-Up)			Non-Acetylsalicylic Acid Antiplatelet Drugs (6 y = Years of Maximal Follow-Up)		
Control diet	38/2191 (1.73%)	1 (Ref.)	1 (Ref.)	48/2191 (2.19%)	1 (Ref.)	1 (Ref.)
MedDiet-EVOO	27/2361 (1.14%)	0.60 [0.36; 0.99] (*p* = 0.045)	0.62 [0.38; 1.04] (*p* = 0.069)	47/2361 (1.99%)	0.78 [0.52; 1.17] (*p* = 0.232)	0.83 [0.54; 1.27] (*p* = 0.389)
MedDiet-Nuts	30/2215 (1.35%)	0.73 [0.46; 1.18] (*p* = 0.200)	0.79 [0.49; 1.28] (*p* = 0.336)	45/2215 (2.03%)	0.86 [0.57; 1.28] (*p* = 0.446)	0.89 [0.59; 1.35] (*p* = 0.579)

Hazard ratios were estimated by multivariable Cox proportional hazards regression models. Model 1 was adjusted for sex and recruitment site as strata variables, and age. Model 2 was further stratified by educational level as strata variable, diabetes, hypercholesterolemia, hypertriglyceridemia, hypertension, smoking habit, leisure-time physical activity, body mass index, alcohol consumption (at baseline); and two propensity scores that used 30 baseline variables to estimate the probability of assignment to each of the intervention groups. We used robust standard errors to account for intra-cluster correlations. MedDiet-EVOO: Mediterranean Diet enriched with extra-virgin olive oil; MedDiet-Nuts: Mediterranean Diet enriched with mixed nuts.

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
