# Peer review of "Mediterranean Diet Decreases the Initiation of Use of Vitamin K Epoxide Reductase Inhibitors and Their Associated Cardiovascular Risk: A Randomized Controlled Trial"

_nutrients, 2020, doi:10.3390/nu12123895_

Round 1

Reviewer 1 Report

Comments to the Author
The paper by, Castro-Barquero et al: Mediterranean diet decreases the initiation of 2 vitamin K antagonist therapy and its associated 3 cardiovascular risk: a randomized controlled trial it’s interesting enough but it needs minor revision.

  • both in the abstract and in the introduction the Authors are strongly advised to describe where VKA act: the vitamin K cycle, Vitamin K1 and K2, Vitamin K Dependent Proteins (VKDPs) especially Bone Gla Protein and Matrix Gla Protein and their beneficial effects on Bone and Vascular Calcifications
  • Page 3, line 112: The Authors are strongly advised to improve Figure 1
  • Page 3, line 115: the Authors are strongly advised to describe what kind of food frequency questionnaire they used. Furthermore the Authors should describe the content of vitamin K in extra-virgin olive oil.
  • Table 1, line 207: The Authors should show p-value for each variable
  • Table 2, line 221: divide the table into three parts according to the use of the following drugs: vitamin K antagonists, acetylsalicylic acid, and other antiplatelet
  • Figures S1: should be shown in the main text
  • Discussion: The Authors are strongly advised to are strongly advised better describe clinical vitamin K intake studies including extra-virgin olive oil with resulting decrease in cardiovascular disease without or with renal failure (the Authors are strongly advised to add as reference Fusaro et al: Low vitamin K1 intake in hemodialysis patients. Clin Nutr. 2017 Apr;36(2):601-607. DOI: 10.1016/j.clnu.2016.04.024. Epub 2016 Apr 28; it is the first study on the Mediterranean diet in hemodialysis patients). Moreover, the Authors should describe if MedDiet enriched in extra-virgin olive oil reduced the risk of initiating treatment with novel anticoagulant drugs (eg rivaroxaban, apixaban).

Author Response

  • Both in the abstract and in the introductionthe Authors are strongly advised to describe where VKA act: the vitamin K cycle, Vitamin K1 and K2, Vitamin K Dependent Proteins (VKDPs) especially Bone Gla Protein and Matrix Gla Protein and their beneficial effects on Bone and Vascular Calcifications.

We agree with the reviewer that the denomination “vitamin K antagonist” was not precise enough. We have substituted it in the whole text for “vitamin K epoxide reductase inhibitors”, which better reflects the pharmacological action of warfarin and acenocoumarol. In addition, there is a more complete description of the biochemical action of vitamin K epoxide reductase inhibitors (lines 81-84 of the manuscript file with tracked changes):

A reduction in the risk of initiating the use of vitamin K epoxide reductase inhibitors (molecules preventing the in vivo regeneration of vitamin K, necessary for the activation –through carboxylation– of coagulation factors II, VII, IX and X which, in turn, decrease the capacity of the organism to trigger the coagulation cascade) […]”

  • Page 3, line 112: The Authors are strongly advised to improve Figure 1

We agree with the reviewer that the exclusion process and Figure 1 were not clearly described. In brief, we assessed the initiation of vitamin K epoxide reductase inhibitors, acetylsalicylic acid, and non-acetylsalicylic acid antiplatelet drugs in individuals with follow-up information on drug use (we excluded 297 participants because of this –51 in the MedDiet-EVOO group, 119 in the MedDiet-Nuts arm, and 127 in the control diet–) and not using these medications at baseline (we, respectively, excluded the 135, 1,136, and 124 users of vitamin K epoxide reductase inhibitors, acetylsalicylic acid, and other antiplatelet drugs at baseline). Users of non-acetylsalicylic acid antiplatelet drugs and vitamin K epoxide reductase inhibitors at baseline were additionally excluded from the analyses on the initiation of use of vitamin K epoxide reductase inhibitors and non-acetylsalicylic acid antiplatelet drugs, respectively (both are considered antithrombotic medications in severe patients and are not generally used combined). Finally, as explained in the Statistical Methods section, any initiation in drug use during the first year of follow-up was excluded to minimize reverse causation.

We have reworded this description in the text (lines 113-126 of the manuscript file with tracked changes) and re-structured the study flow chart to clarify the creation of our analytical samples.

  • Page 3, line 115: the Authors are strongly advised to describe what kind of food frequency questionnaire they used. Furthermore the Authors should describe the content of vitamin K in extra-virgin olive oil.

We have now clarified that we used a 137-item, semi-quantitative food frequency questionnaire validated in Spanish adult population. It was referenced in Citation #13, we have rearranged this sentence in order to clarify the description (lines 170-174 of the manuscript file with tracked changes). As a point of clarification, we transformed the questionnaire results into the estimated daily servings of each alcoholic beverage (different kinds of wines, beers, and spirits) and, finally, computed the estimated amount of ethanol consumed by each participant in grams/day.

Regarding the content of vitamin K in extra-virgin olive oil, we have only been able to find this information in two publicly available records of the US Department of Agriculture database (https://fdc.nal.usda.gov/fdc-app.html#/food-details/1103861/nutrients; https://fdc.nal.usda.gov/fdc-app.html#/food-details/171413/nutrients). Both of them referred to olive oil (no distinction between virgin or refined oils) and agreed to report that this product contains 60.2 μg/100 g. This implies that each 10 g spoonful of oil provides 5.0 and 6.8% of daily Adequate Intakes of vitamin K in adult men and women, respectively, according to the Institute of Medicine (https://www.ncbi.nlm.nih.gov/books/NBK222310/).

This information is now available in lines 144-147 of the manuscript file with tracked changes, and we have included the previous citations as new references:

It must be pointed out that olive oil is a remarkable dietary source of vitamin K1 (phylloquinone), providing 60.2 μg/100 g of oil (which implies that each 10 g spoonful of oil provides 5 and 6.8% of daily Adequate Intakes of vitamin K in adult men and women, respectively, according to the Institute of Medicine)”.

  • Table 1, line 207: The Authors should show p-value for each variable

P-values have been systematically added after every hazard ratio and its 95% CI.

  • Table 2, line 221:divide the table into three parts according to the use of the following drugs: vitamin K antagonists, acetylsalicylic acid, and other antiplatelet

Following the reviewer’s suggestion, Table 2 has been split into three independent tables for vitamin K epoxide reductase inhibitors (new Table 2), acetylsalicylic acid as antiplatelet drug (Table 3), and non-acetylsalicylic acid antiplatelet drugs (Table 4). We have also included the analyses on onset of use of vitamin K epoxide reductase inhibitors after excluding prevalent and incident cases of atrial fibrillation.

  • Figures S1: should be shown in the main text

Figure S1 is now available in the main text as Figure 2.

  • Discussion:The Authors are strongly advised to better describe clinical vitamin K intake studies including extra-virgin olive oil with resulting decrease in cardiovascular disease without or with renal failure (the Authors are strongly advised to add as reference Fusaro et al: Low vitamin K1 intake in hemodialysis patients. Clin Nutr. 2017 Apr;36(2):601-607. DOI: 1016/j.clnu.2016.04.024. Epub 2016 Apr 28; it is the first study on the Mediterranean diet in hemodialysis patients). Moreover, the Authors should describe if MedDiet enriched in extra-virgin olive oil reduced the risk of initiating treatment with novel anticoagulant drugs (eg rivaroxaban, apixaban).

Following the reviewer’s suggestion, we have added to the Discussion a mention of previous vitamin K intake studies within a Mediterranean dietary pattern and cited the suggested reference (lines 325-328 of the manuscript file with tracked changes):

The role of vitamin K in cardiovascular mechanisms and the intake of vitamin K within a frail population following a MedDiet has been previously discussed (Fusaro M et al., Clin Nutr, 2017) […]”.

Regarding the second comment, we were unable to study the changes in the risk of initiating the use of more advance oral anticoagulant therapies such as direct oral anticoagulants. As described in the methodology, our participants were recruited between 2003 and 2009 and followed until December 1, 2010 at the latest. During the study follow-up, the use of direct oral anticoagulants was extremely scarce and thus no changes in the risk of initiating this kind of therapy could be assessed. We have added this aspect as a Limitation (lines 357-360 of the manuscript file with tracked changes):

“Second, the use of more modern antithrombotic therapies (e.g. direct oral anticoagulants such as dabigatran, apixaban, rivaroxaban, and edoxaban) was extremely scarce during our study follow-up (2003-2010) and we were unable to investigate changes in the risk of initiating the use of these medications”.

Reviewer 2 Report

This is a post hoc analysis of a large randomised trial comparing Mediterranean Diets with 1) olive oil or 2) nuts with a low fat control diet.

The authors have explored

1) if the diets influenced on the risk of initiating vitamin K antagonists, acetylsalicylic acids or other antiplatelet drugs

2) if the diets affected the risk of nonfatal coronary heart disease, nonfatal stroke or death in patients treated with vitamin K antagonists, acetylsalicylic acids or other antiplatelets at baseline

I think the authors are a bit too optimistic when considering the results:

  1. The main result is a reduction in initiation of VKA in the MedDietEVOO: 0.68[0.46;0.998] Why three decimals in the upper confidence interval boundary? This is the only result where three decimals are provided. Perhaps because this was necesary in order to demonstrate significance?
  2. The absolute risk reduction was less than 1%. I suggest these numbers are provided to the reader.
  3. The p-value for interaction betweeen VKA and diet did not show signifcance

Other remarks:

  1. Could a possible protective effect of MedDietEVOO be due to protection against hemorhagic stroke casued by the the content of vitamin K in the oil?
  2. How was the effect in the group with atrial fibrillation?

Author Response

  • The main result is a reduction in initiation of VKA in the MedDietEVOO: 0.68[0.46;0.998]. Why three decimals in the upper confidence interval boundary? This is the only result where three decimals are provided. Perhaps because this was necessary in order to demonstrate significance?

We agree with the reviewer that this particular result was significant by a narrow margin. Following the suggestions of Reviewer #1, we have added P-values throughout the text and it can be observed that this association was significant according to its 95% CI and its P-value (P=0.049). We used this strategy because the other option to highlight this aspect (to indicate that the upper limit of the confidence interval was “<1.00”) was less aesthetic. However, this finding does not seem spurious since the association between the MedDiet-EVOO intervention and the risk of initiation of VKA therapy was clearly significant in the model considering age/sex/recruitment site as covariates (HR: 0.64 [95% CI: 0.43; 0.94], P=0.021), as well as in the analyses excluding atrial fibrillation prevalent and incident cases (Model 1: HR 0.50 [95% CI: 0.30; 0.83], P=0.008; Model 2: HR 0.53 [95% CI: 0.32; 0.88], P=0.014).

In any case, we agree with the reviewer that we could comment some of our findings in a less bold tone considering this condition. We have therefore eliminated the generalization of the relevance of our results in: 1) the first paragraph of the Discussion (lines 310-312 of the manuscript file with tracked changes: “This is the first study to describe the long-term protective effects of MedDiet on the risk of initiating antithrombotic therapy in an older population at high cardiovascular risk”); and 2) the Conclusions (lines 379-381 of the manuscript file with tracked changes: “To the best of our knowledge, this is the first study to describe the long-term protective effects of MedDiet on the risk of starting to use antithrombotic therapies”).

  • The absolute risk reduction was less than 1%. I suggest these numbers are provided to the reader.

Following the reviewer’s comment, we have added this information in the Results section (lines 235-236 of the manuscript file with tracked changes):

“The difference between the incidence rate in the control diet and the MedDiet-EVOO intervention was 0.66%”.

  • The p-value for interaction between VKA and diet did not show significance.

We agree with the reviewer that this P-value was only marginally significant (P=0.052). However, we have not focused in the statistical significance of the P-value when we have commented the results of the interaction between the use of VKA at baseline, allocation to MedDiet, and major adverse cardiovascular events (MACE). As stated in the Abstract and other parts of the text, we only highlight that VKA use was more strongly associated with an increased risk of cardiovascular disease in participants not allocated to MedDiet interventions. First, the association between VKA use at baseline and incidence of MACE was very strong in the participants allocated to the control group (HRcontrol diet: 4.22 [1.92-9.30], P<0.001), whilst the association between VKA therapy and MACE incidence in participants allocated to the MedDiet interventions was not significant (HRMedDiets: 1.71 [0.83-3.52], P=0.137). Second, the magnitude of the association was attenuated in a very considerable manner: participants in the control group presented a 322% excess risk of developing a MACE, whilst those in the MedDiet groups presented a non-significant 71% increase. Finally, regarding the interpretation of interaction tests, several publications allow the consideration of marginally significant P-interaction values (P-value <0.1) due to the extremely demanding nature of this type of analysis. Two examples of high-profile publications using this strategy are the following: Beddhu S et al., Circulation, 2018; Soria-Florido MT et al., Circulation, 2020. Considering all the previous arguments, we should not rule out there is a differential association between VKA use and MACE risk depending on the intervention group.

However, following this comment we have added this aspect as a Limitation to our work (lines 366-373 of the manuscript file with tracked changes):

“Finally, the P-value for the interaction between the use of vitamin K epoxide reductase inhibitors at baseline and the intervention group on the risk of developing a major adverse cardiovascular event was only marginally significant (P=0.052). However, we decided to interpret the interaction since the association between drug use and cardiovascular incidence was very strong in the participants allocated to the control group but non-significant (and of a very lower magnitude) in those allocated to MedDiet, and several previous publications have allowed the consideration of marginally significant P-interaction values (P<0.1) due to the extremely demanding nature of this type of analyses (Beddhu S et al., Circulation, 2018; Soria-Florido MT et al., Circulation, 2020)”.

Other remarks:

  • Could a possible protective effect of MedDietEVOO be due to protection against hemorrhagic stroke caused by the content of vitamin K in the oil?

Following this comment and others from Reviewer #1, we have included in the manuscript some information on the content of vitamin K in MedDiet foods such as extra-virgin olive oil and its potential effects on cardiovascular health in some populations at high cardiovascular risk (lines 144-147 and 325-328 of the manuscript file with tracked changes).

  • How was the effect in the group with atrial fibrillation?

As can be observed in the following results, no changes in the risk of initiation of VKA therapy in atrial fibrillation patients were observed (although the incidence rates of VKA initiation were considerably higher in this population, as expected):

Cases/Total

(Incidence rate)

Model 1

(HR [95% CI])

Model 2

(HR [95% CI])

Control diet

22/57

(38.6%)

1 (Ref.)

1 (Ref.)

MedDiet-EVOO

24/55

(43.6%)

0.91 [0.52; 1.58]

(P=0.731)

1.18 [0.63; 2.20]

(P=0.601)

MedDiet-Nuts

23/53

(43.4%)

0.75 [0.43; 1.31]

(P=0.311)

0.70 [0.35; 1.40]

(P=0.315)

Round 2

Reviewer 2 Report

Thank you for the reply. I have no further comments